# “Unworthy of Care and Treatment”: Cultural Devaluation and Structural Constraints to Healthcare-Seeking for Older People in Rural China

**DOI:** 10.3390/ijerph17062132

**Published:** 2020-03-23

**Authors:** Xiang Zou, Ruth Fitzgerald, Jing-Bao Nie

**Affiliations:** 1Department of Medical Humanities, Southeast University, Nanjing 211189, China; 2Department of Anthropology and Archaeology, University of Otago, Dunedin 9016, New Zealand; ruth.fitzgerald@otago.ac.nz; 3Bioethics Centre, University of Otago, Dunedin 9016, New Zealand; jing-bao.nie@otago.ac.nz

**Keywords:** health-seeking, older people, structural constraints, cultural devaluation, rural China

## Abstract

This paper examines the experiences of seeking healthcare for rural Chinese older people, a population who experiences the multiple threats of socio-economic deprivation, marginalization, and lack of access to medical care, yet have been relatively overlooked within the existing scholarly literature. Based on ethnographical data collected from six-month fieldwork conducted in a rural primary hospital in Southern China, this paper identifies a widespread discouraging, dispiriting attitude regarding healthcare-seeking for rural older members despite the ongoing efforts of institutional reforms with a particular focus on addressing access to health services amongst rural populations. Such an attitude was expressed by older people’s families as well as the public in their narratives by devaluing older members’ health care demands as “unworthy of care and treatment” (“*buzhide zhi*” in Chinese). It was also internalized by older people, based on which they deployed a family-oriented health-seeking model and strategically downgraded their expectation on receiving medical care. Moreover, underpinning this discouragement and devaluation, as well as making them culturally legitimate, is the social expectation of rural older people to be enduring and restrained with health-seeking. Simultaneously, this paper highlights the sourc2e of institutional and structural impediments, as they intersect with unfavorable socio-cultural values that normalize discouragement and devaluation.

## 1. Introduction

With a rapidly aging global population, there is an urgent need to address barriers older people face in accessing health care so as to promote proactive health-seeking behaviors amongst them [1]. Such need is singularly pressing in Chinese rural regions, where older members, though over half of them are living with chronic symptoms and other co-morbidities, are deprived of access to necessary health and long-term care services, such as chronic care services and medical treatments, under a persistent rural–urban disparity [2,3,4]. In a survey conducted in 2014, close to 31% of rural residents did not seek outpatient services, and 7.5% did not utilize hospital care services when necessary [3]. A persistent rural–urban regional inequality in the distribution of health care resources has directly undermined rural older people’s access to necessary health care [3,5]. Data from two surveys conducted by the Chinese Ministry of Health and Family Planning in 2015 show that only 33% of China’s total medical care resources were allocated to rural areas, even though the rural population constitutes over half of China’s entire population [6]. The Chinese government has been steadily reforming the health system with a focus on rebuilding rural primary care and reverse inequalities. However, recent empirical work suggests that reformative efforts at the institutional level have not resulted in an obvious increase in medical care utilization and health-seeking behaviors amongst rural populations [4,7]. Indeed, the rural–urban health disparity, as reported by another study, has grown [8]. This paradox suggests the need for more research on what factors and constraints discourage the health-seeking behaviors of rural Chinese older populations. Despite these rural older members struggling with health care, their lived voices have been largely overlooked in existing literature since there seems to be an assumption that China should have no problem with caring for older people’s dependent needs because of Chinese Confucian teaching that emphasizes revering older people and the younger generation’s filial obligation to care for the aged [9,10]. An empirical exploration of the experiences of the health-seeking of older people in rural China also has the potential to rectify this idealized assumption. 

### 1.1. Backdrop

The practice of health-seeking of rural older people is located in a complex social, institutional, and structural backdrop that has exposed them to more hardships and obstacles in accessing medical care services in China. The trend of the commodification of the Chinese health care industry since the reforms of the market economy in the 1980s is a major institutional impediment to access medical care for most rural residents [11,12]. In general, the Chinese health system experienced a major transformation that can be roughly divided into two phases: a collective-based health system pre-1980 and a market-dominated health care industry post-1980s. Since the establishment of the People’s Republic of China in 1949, a three-tier health system has gradually been established, with a community-funded Cooperative Medical Scheme (“*hezuo yiliao*” in Chinese) at the rural primary level to guarantee basic access to medical care of over 90% of rural dwellers [13,14]. Most rural residents enjoy accessible health care services provided by “barefoot doctors” [14], who offer health services to local communities while continuing farming vocations free of charge, although the health care provided might be less comprehensive than services in urban centers. 

However, as the free market reforms since 1980s have led to the disintegration of the collective economy, the government has gradually retreated its funding for social welfare systems such as health care and education; hence, Chinese public hospitals have become under-funded and are motived to be financially self-reliant through self-enterprise. The coming decade has seen a radical trend in for-profit entrepreneurship under which public hospitals and health providers have become increasingly reliant on the revenues generated by fees for high-tech intervention and excessive prescriptive behaviors [5,11,12]. Consequently, the rapid increase in medical costs has further disadvantaged most rural older people who are typically of lower socio-economic status [15,16,17]. Studies conducted in 2014 reported that 14.4% of rural households endured catastrophic health expenditure and another 9.2% experienced medical impoverishment, more than those in urban China [2,18]. Also, the insurance coverage rate of rural patients has been compromised by market reforms. Indeed, until the end of the 1990s, over 95% of the rural population were deprived of any insurance coverage and had to shoulder medical expenses through their own financial means [1].

Having realized the threats posed by rural–urban health disparities to social stability and economic development in the long term, the Chinese state embarked on a series of health system reforms with a focus on rebuilding rural primary care to reverse regional inequalities. For example, in response to the need for affordable medical care, a universal health insurance coverage was gradually established from 2002 to 2009, with over 90% of the rural populations insured under the New Rural Cooperative Medical Scheme (NRCMS) [19]. In addition, the Zero Mark-up Prescriptions drug policy has been implemented to restrict health providers to only prescribe drugs from the National Essential Drug List [12] to reduce previous inappropriate incentives for health professionals to overprescribe by eliminating mark-ups on medications. However, the extent to which these reforms have alleviated rural residents’ financial burdens and promoted their health-seeking behavior remains unclear. An empirical study reported an increase of 54.7% per capital annual health expenditure from 2004 to 2010, and the post-reform rural medical scheme has not prevented catastrophic health expenditures in households with chronic diseases [2]. Another study highlighted that the new drug policy, as long as inhibiting the over-prescription behaviors of health professionals, also restricted their provision of specialist services and clinical autonomy. Consequently, some rural residents found it even more difficult to access quality medical care at the rural primary level, and, according to certain criteria, it has been reported that health disparities have grown since the reforms [20,21].

The growing inequality in accessing medical care services between rural and urban dwellers was further intensified by the *hukou* system(the Chinese household registration policy) [22,23], which was signed into law in 1958. As a central stratifying determinant of the unequal access to medical care between rural and urban residents, the *hukou* system assigns all residents to one of two types of households, rural or urban, as well as designated the type of employment that residents were entitled to. Rural residents were subsequently granted inferior access to state-provided social welfare resources. For example, urban residents were entitled to relatively better-quality health insurance coverage that offered them a higher rate of coverage when seeking medical care services in large hospitals in urban areas. Due to the restricted scope of health insurance coverage regulated under the *hukou* divide, rural residents seeking health care are mainly restricted to the rural primary care level and face lower rates of insurance coverage (in contrast to urban dwellers), and so are much more out-of-pocket when seeking health care at upper-level hospitals. 

The *hukou* policy also disadvantages rural older people’s financial means to medical care by excluding them from equal access to the social pension system, as only urban residents have access to formal employment and, hence, employment-based pensions after their retirement. Rural residents, who mainly work in the agricultural sector, must rely on their personal savings and families to secure late-age care and support. This *hukou*-based structural divide means that rural residents aged 60 years and older face unjust marginalization in accessing healthcare services. Additionally, this system has shaped the way rural residents perceive their social status and their access to national welfare. Existing studies identified a strong sense of inferiority amongst rural residents when compared to their urban counterparts, a sense which was linked to the unequal rural–urban division of social welfare resources. For instance, there is a derogatory definition of rural villagers as peasants and “bad” citizens who are uncivilized and immoral [24], and as “second-class”, “backward peasants” who have no right to expect support from the state [4,25].

In the face of the institutional plight of *hukou* constraints and the dearth of formal support, most rural older people have no choice but to rely on their families to secure means to necessary medical care services. Under the influence of the Confucian norm of filial piety, families have long been the primary institutions responsible for the later-age security of older members in China [9,26]. In the context of seeking health care of older people, families are expected to manage their medical care expenses, accompany them on hospital visits, and navigate hospital care services. However, this traditional model of family-based healthcare-seeking for older people has come under pressure since Chinese society experienced modernization and marketization. The first challenge arose from the massive outward migration of the younger generation. Since the initiation of market economy reforms of the 1980s, the development of urbanization has attracted numerous rural younger people to migrate away from rural areas to urban areas to seek employment, thereby undermining the human resources and capabilities of individual rural families to assist in healthcare-seeking when their older members fall sick [27,28].

Additionally, the trend of modernization and marketization has also altered the way people perceive the meaning of later-age care and support. Traditionally, most older people enjoyed a generally high social and family status and were honored for their mastery of history knowledge and wisdom as well as representation of high moral standards [29]. As regulated by the norm of filial piety, the younger generation should respect and care for older members, which was an essential, ubiquitous practice. In contrast, throughout the economic reforms in the past three decades, marketization and privatization have engendered a new image of self-reliant and independent individuals who are capable of bearing their own responsibilities for welfare and interests [30,31], such as late-age support and health care. Under this self-reliant social discourse, the younger generation increasingly perceives themselves as individualized against parental authority, attending to the pursuit of personal interests and desires over the collective commitment of care provision [32]. This self-reliant social discourse is also problematic for the image of most older people who are dependent and senile, relying on external support to survive [33], an image that is incompatible with the character of competitiveness and profitability endorsed under the market economy. This preconception is pervasive, despite the fact that most rural people were already at least partially reliant on their families and themselves own for their welfare prior to market economy reforms [34]. Both individual families and the state are reluctant to support aging health care, with some rural older people having even internalized the idea of their healthcare needs as “burdensome”, adopting passive attitudes in medical care services [4]. 

### 1.2. Conceptual Framework

Existing literature on health-seeking behavior suggests two analytical lenses to understand the low utilization of health care services amongst Chinese older rural populations. In Yan Long’s and Lydia Li’s studies on patients’ health-seeking behaviors in China, they summarize the various forces, which can be classified as structure-constrained and culture-determined, that contribute to a rural–urban dichotomy of health-seeking behaviors [4]. Informed by the political–economic theories of inequalities, the structure-oriented branch of literature emphasizes the impacts of macrostructural constraints and the institutional organization that contribute to financial barriers in accessing to health care services within given populations [11,12,15,16]. This body of literature aimed to describe certain populations’ worse-off health performance at the statistical level and was produced through policy development studies. Consequently, the question regarding how these macro constraints operate in rural older people’s health-seeking decisions and related behaviors at the micro level is devoid of proper empirical examination. 

In contrast, the culture-determined branch of literature calls for more interpretative approaches to social and cultural values in explaining patients’ beliefs and choices of health-seeking and disease management [4,35,36]. This literature rests the analysis at micro, interpersonal, and familial levels, being more concerned with meaning–interpretation and illness narratives of individual patients. Some scholars have demonstrated, while the accessibility of external healthcare resources matters, how factors such as individual health literacy, family beliefs, and cultural values deeply influence the way individuals respond to chronic illness and manage health care in the act [4,37,38,39,40]. This approach of literature emphasizes the importance of local social conditions and cultural values to study health-seeking behaviors of certain groups of populations. Those local conditions and values are already in place, which, in turn, impede or facilitate people’s decisions regarding whether and how to engage healthcare systems [38,41]. 

A small but growing body of literature has attempted to mitigate the divide between these two branches of literature [38,41,42,43,44] by proposing a relational account [4] of analysis of health-seeking behaviors. The structural and cultural facets intertwine in a way that produces inequalities, as coined in Long’s work, “deeply imbued with legitimizing rationales wherein present prevailing arrangements are perceived as natural” [4] (p. 3). In this regard, Lora-Wainwright’s work on caring for cancer patients in rural China offers a very good example: she illustrates the contrast between the economic hardship faced by rural households and the commodification of the health care industry, tracing the ways in which these phenomena shape financial barriers to access diagnostic services and care and, ultimately, how this forces cancer patients to delay diagnosis and treatment. Additionally, she offers a local cultural interpretation on people’s refusal of cancer treatment and care: the public’s attitude towards cancer as a type of incurable disease has produced the perception of treatment as a waste of resources, making their refusal of treatment socially legitimized [38] (pp. 19–20). 

Another important level of analysis informed by the relational account is at the engagement of families in health-seeking and chronic care management of patients. This is especially true for those older people living in Chinese rural regions, where, as introduced above, the constraint of *hukou* and the rural–urban divide have exacerbated the financial and physical barriers rural populations encounter in accessing to a market-dominated health care industry and, in turn, shifting the primary burden of healthcare of older people onto individual families. Due to the process of health-seeking of older members, both older people and their families have come to a shared understanding of health-seeking as a collective family event. The decisions regarding whether and how to engage in health-seeking is made by considering the interests of the whole extended family, rather than based on the autonomous choices of older individuals. In this regard, Moreira, through her study on caring for Alzheimer patients in Britain, argues what is at stake regarding healthcare for Alzheimer patients: it is not staged as a fight against the inevitable cognitive decline, but rather the collective events of family caregiving [45]. Lora-Wainwright’s ethnographic work also offers a strong example depicting how, in the face of the unaffordable medical expenses, patients in China resort to home-based care rather than reliance on medical institutions. 

Both structural-focused and cultural-centered approaches are informative to analyze the health-seeking behavior of rural older populations in the Chinese context. The theoretical proposition of this paper strives to demonstrate the dynamic interplays between these two approaches and the importance of developing a family-oriented health-seeking model to understand the various accounts of values and beliefs related to the meaning of aging care, chronic illness experiences, and health-seeking behavior. These values and beliefs are embedded in older people’s daily experiences and family relations, based on which rural older people (and their families) have adopted specific ways in their evaluation of healthcare and health-seeking management. By understanding health-seeking behavior in a relational term and as determined by multidimensional factors, this paper partly demonstrates why China’s current health institutional reforms in themselves might not be sufficient to produce proactive health-seeking behavior due to the durable influence of the local cultural system and its intersection with social structures that discourage health-seeking, although these socio-cultural determinants have been challenged by a set of reformative health policies [8]. It is also informative to follow-up on national health reform in addressing these cultural and structural impediments and accomplishing its goals in promoting health equity. 

## 2. Methods and Materials 

This paper juxtaposes the narratives of older patients and their families to understand the processes underlying seeking health care of older people in rural China. It employs ethnographic methods, comprising participant observations, semi-structured interviews, and unstructured interviews, to document the rural older patients’ and their respective family caregivers’ narratives and experiences of seeking medical care and treatment. The fieldwork was conducted between January and June 2016 in a rural primary hospital, Qincun Hospital, in Guangdong Province, China. Qincun Hospital is located in Qincun town, a rural locality that is financially impoverished and geographically isolated. It is also of a relatively moderate size and scale, which could better facilitate the participation of medical staff and patients in the observational work and interviews conducted for the study. According to the official 2015 census records, Qincun Hospital had 45 certified physicians, six general practitioners, and 21 nurses providing active service. In the inpatient ward, where most of the field work took place, there were 50 inpatient beds available, with 20 physicians and 10 nurses working together to manage the major tasks of inpatient care delivery. The six-month ethnographic fieldwork and all the interviews were carried out by the lead author (Zou), an anthropologist–bioethicist. Throughout the field work, 20 older inpatients and their family caregivers, ten medical staff, and other overlapping parties were interviewed. All interviewed patients were over 60 years old and most of them suffered from multiple chronic health problems: 13 of 20 patients were enrolled due to cardiovascular disease and dementia; another four patients suffered from advanced cancer or other terminally ill conditions. The length of time that the interviewed patients spent in Qincun Hospital varied, from five days up to two months. 

During the current study, most local young people had migrated away from Qincun to other affluent urban areas nearby, such as the Pearl River Delta, to seek employment, with the majority of Qincun’s population being those left-behind older people. Additionally, as most older people had relatively high levels of healthcare demands, they were more likely to seek hospital care services. Although official statistics regarding the age distribution of patients at Qincun Hospital were unavailable, nearly 90% of patients encountered and/or interviewed were at least 60 years of age; all were from the Qincun locality. The older demographic trend of enrolled patients combined with their high-demand, chronic care needs has somehow transformed the primary function of Qincun Hospital from a medical institution aimed at medical care interventions to a gerontological care facility targeted towards meeting the chronic and palliative care needs of older dwellers.

The study sample was focused on rural older inpatients due to the high-demand healthcare and the need of family support in the context of hospitalization. This is because the Chinese health system only trains nursing staff to focus primarily on conducting technique-oriented nursing care interventions; therefore, a set of inpatient bedside care tasks, including feeding, toileting, changing patient positions, and maintaining daily hygiene, ultimately falls on patients’ families [46]. This intensive nature of nursing care needs has made hospitalization an ideal site for an investigation of the family-oriented health-seeking practice of older people in rural China. 

Financial hardship is a major obstacle to the interviewees’ access to medical care resources. Despite that all of the interviewed patients were insured under the NRCMS, the actual reimbursements they received from NRCMS were restricted to only a portion of medical care services, as many services, such as nursing services and the fee for an inpatient bed, are not covered by the NRCMS. The reimbursement mechanisms of NRCMS are also complex. According to the NRCMS regulation in Qincun locality, the coverage rate under NRCMS could reach up to over 90% when patients utilized inpatient care services but may decrease to 50% with utilizing outpatient services and less than 30% when seeking medical care in other tertiary hospitals. This means the out-of-pocket expenses remain unaffordable for most older patients, who are simultaneously deprived of access to formal social pension systems due to their rural *hukou* status.

Data collection began with informal interviews to establish good rapport and trust with older patients and their families. In the case of each patient, one semi-structured interview with three to four follow-up unstructured interviews were conducted. Interviews mainly focused on patients’ health conditions, their previous health-seeking experiences, financial sources for inpatient treatment, and their experiences with, and perceptions of, their chronic symptoms, health-seeking, and care-receiving. Questions also inquired about family relationships and shared family ties between older patients and their family caregivers, current living arrangements and source of aging support, and their personal life history. Meanwhile, the voices and storytelling of other overlapping parties, i.e., doctors, nurses, other patients, and their families, were also taken into account to formulate a nuanced picture of supporting health-seeking and caregiving in a given family context. All the interviews were conducted in and transcribed to Chinese, while ethnographic notes were taken by the first author. All identifying information was removed to protect the confidentiality of research participants and all names used in this paper are pseudonyms. Informed consent was given by all participants involved in this study. Ethical approval was obtained from the Human Ethics Committee of the University of Otago in New Zealand (Reference No. 15/106). 

This study adopts an abductive approach [47] to analyze the qualitative data with NVivo software (QSR International, Doncaster, Australia) sourced from New Zealand. Initially, inductive coding techniques were applied to discern a set of themes, patterns, and trends, as well as subthemes and overarching raised by interviewees. Then, the analysis moved back and forth between coding themes and conceptual frameworks, focusing on the interplays between social structures and cultural facets to develop an explanatory model of health-seeking behaviors of rural older Chinese patients. This study particularly focuses on identifying the influences of the local system and cultural values as they are ingrained into the lives of individual interviewees. 

## 3. Results

The structural reality of socio-economic deprivation and lack of institutional support do not act alone in producing the low utilization rate of medical care amongst the rural older populations. Throughout the interviews, a series of unfavorable values were identified that discourage older members’ health-seeking by devaluing rural older people’s healthcare as “unworthy of care and treatment”. As elaborated below, this “worthlessness” (“*buzhi*” in Chinese) discourse is suggested in the way rural families, as well as medical professionals, perceive the significance of older members’ health and respond to medical decisions. A sense of “worthlessness” was also internalized by many older patients, based on which they strategically normalized chronic symptoms and actively avoided utilizing hospital care, although a few of them also expressed a repressed emotion against discouragement and desire for medical care. In addition, the local moral belief that encouraging older people to endure chronic pain and suffering compounds public discouragement and the perception of older members’ proactive health-seeking behaviors as culturally “inappropriate”. The analysis also elaborates on how these unfavorable attitudes operate as they intersect with a macro-structural deficiency to normalize the prevalence of devaluation and decimation. 

### 3.1. “Unworthy of Care and Treatment”: Public Devaluation Regarding Seeking Health Care of Older Members in Qincun Hospital

In the face of the lack of access to quality medical care and gerontological care facilities, most of which were located in urban areas, Qincun Hospital was sought as the most convenient solution to older people’s health and chronic care demands. Coupled with a high reimbursement rate of 90% achieved through the NRCMS under the condition of utilizing inpatient care services, local dwellers have become increasingly willing to utilize inpatient care at Qincun Hospital. However, despite this, most dwellers’ attitudes towards doctors’ motivations and the quality of care provided in Qincun Hospital were skeptical. For instance, a daughter-in-law who accompanied her mother-in-law commented that doctors from Qincun Hospital “(knew) nothing about treatment, but only selling fake medication”. Regardless of this low expectation, she asserted Qincun Hospital was “sufficient” for her 94-year-old mother-in-law because she was “too old to be healed*”.* This low expectation was partly due to the incurability of her mother-in-law’s chronic symptom, as she described it in terms of “too old to be healed”. It also conveyed a trend of devaluation perceived by local people that the health of older family members was not as worthy of medical care and attention. This daughter-in-law’s opinion was not an exception. Additionally, some medical staff expressed a sense of discouragement and devaluation, asserting it was “unwise” to devote too much medical care and attention to older patients. The opinion of a doctor in Qincun Hospital was particularly typical: 


*“At their (older people’s) age, peers around them have passed away one by one … How can they expect more? It (their health) is not worth treating that seriously. They know they will never be able to get well, and it is unwise to devote too much to their medical care.”*


The voices and desires of most rural older people were also overlooked when they relied extensively on their families to secure financial and physical means to hospital care services. In most of the families interviewed, older people were deprived of power and autonomy to make decisions regarding their medical care and treatment. Instead, they were typically described as backward, out-of-date and “knowing nothing” by their families, and therefore not capable of making own decisions or even just expressing their own opinions. For example, there was an instance when a health professional suggested that the researcher to interview the patients’ families, rather than the older patients: “These older patients know nothing about their disease or treatment. You should talk to their families about their health-seeking experiences”. In another example, a 66-year-old patient who suffered from brain atrophy described his symptoms and asked the researcher with a serious tone, “Do you know if there is any technology that can, as such, open up my brain, shake it well, and stop it from shrinking?” But his family member thought him ridiculous and attempted to stop him as well as the researcher from talking to the patient: “Stop your insane words! You know nothing! There is no use to talk to him (the older person)!” 

The trend of devaluation regarding the significance of elderly health-seeking was pervasive. Along with the constrained financial situation of most rural families, devaluation contributed to the decision to attend Qincun Hospital as the most practical resolution. When admitted to Qincun Hospital, most interviewed rural dwellers attempted to postpone older members’ treatments by relieving their chronic symptoms and minimizing the financial burden of the whole extended family, rather than seeking quality care and other effective treatments to extend their lifespan. “I know I cannot be cured here. But I felt so horrible without these (intravenous) drips, like I would have died …”, said an older inpatient who has frequently been admitted and discharged from Qincun Hospital. In the face of the incurability of older people’s chronic symptoms and public discouragement, hospitalization and health-seeking are not staged as a fight against chronic diseases, but rather a process regarding how to endure pain and suffering from one day to the other. The following analysis introduces how older patients responded to public devaluation while managing health care. 

### 3.2. Internalized Discouragement and Subordination

Most interviewed older patients had internalized this “worthlessness” discourse, based on which they strategically reasoned their chronic diseases, postponing hospital care and necessary treatment. Despite the presence of chronic symptoms, they would consider seeking hospital treatment and professional consultation only when their symptoms progressed to a certain level that they perceived as “severe” and difficult. Even when a professional diagnosis was offered, many older people paid little attention to it or adhered to the follow-up medications. For example, a 62-year-old participant who suffered a stroke had fainted on his farm, the doctor recommended referring him to a larger hospital in Yangjiang to seek follow-up rehabilitation, but because the patient was still able to move around, he assumed his condition was minor and refused the doctor’s advice for treatment and proper care. Instead, he only took some medication bought from a pharmacy in his village, on and off, while continuing to work on the farmland until he had another stroke and gradually became paralyzed, so he was admitted for inpatient care in Qincun Hospital at the time of the current study. 

Another older female patient who struggled with multiple chronic diseases expressed a discouraged attitude regarding seeking hospital care and treatment. She usually took large doses of medication to manage her chronic conditions, but they did not work all the time. Only when she was too sick to stand still would she ask her son to send her to hospital for inpatient care, but as soon as she felt a bit better from the inpatient treatment, she insisted on being discharged and reasoned her insistence in this way: “If you can eat and walk, you don’t go to the hospital (“*nengchi nengdong, jiu buyong qu yiyuan*” in Chinese)”. The ability to move and eat was the primary parameter in the evaluation regarding how concerned older members should be with seeking health care. Instead of viewing chronic diseases as acute symptoms with a profound impact on patients’ personal lives, which therefore must be handled immediately, rural older people tended to normalize them as the inevitable consequence of aging, as questioned by another older patient: “Which older person does not have some pain? Does it really need that much treatment?”. 

Furthermore, the discouraging attitudes were normalized, even in the absence of their families assisting with bedside care of older inpatients, of which older members attempted to be self-restrained, downplaying their expectations on family care and support. Due to the insufficient nursing care provided by Qincun Hospital, as aforementioned, the families play an indispensable role in assisting in a set of nursing care activities during older people’s hospitalizations. However, the fieldwork witnessed many family cases in which the commitment to provide basic care needs had been overlooked. There was an older mother, who was in severe chronic pain, who struggled to manage her inpatient care routines (such as three meals) by herself and was virtually “abandoned” by her family after her hospitalization. Nonetheless, her attitude towards the absence of family caregiving during hospitalization was positive: “My family has treated me well enough! Everyone is busy earning money. No work, no income. If my family stopped working and came over to look after me, they would lose their income. Then how could they support my treatment?” 

Older patients’ compliance towards devaluation and adaptive strategies, such as normalizing chronic experiences, being self-restrained with seeking hospital care, and avoiding burdening their families with supporting inpatient care, enabled many rural families to survive, an end which is highly altruistic and noble from the perspective of many local people. “These older people have been through too much hardship over the course of their entire lives (such as poverty and famine). You have no idea how good they are at enduring suffering! They are beyond my expectation”, an interviewed doctor commented in praise of the good endurance of older members. As reported in existing anthropological literature regarding local morality, personal characteristics such as being good at enduring suffering and hardship (“*chiku nailao*” in Chinese) are usually perceived as morally “upright” in Chinese context [37,38]. Older people were expected to maintain thrifty and self-constrained lifestyles, and to be conscious of reducing adult children’s financial burdens, so as to become socially recognized virtuous members, a view that justifies the trend of devaluation and encourages more and more older people to be discouraged with health-seeking.

### 3.3. Older People’s Resistance and Public Accusation

However, not all patients were passive in the same way. During the study, there were some cases in which older people displayed a radical resistance against discouragement, revealing a strong will to pursue healthcare and longevity. There was an 83-year-old mother who suffered from multiple chronic symptoms and was heavily reliant on inpatient care and treatment for over a decade. The whole family felt discouraged from continuing support for her medical care, especially after all three of her children had migrated to cities for employment. For example, there was an occasion that she requested a visit to the hospital for a recurrence of a coronary heart problem, but her family rejected her request since, at that time, the whole family was involved in the preparations for a young family member’s wedding ceremony, asserting that nobody had the time or energy to accompany her to hospital. Seeing there was no way out, the mother threatened to commit suicide during the wedding if her family still rejected her request for a hospital visit: “Don’t be busy with the wedding, prepare for my funeral”. The mother’s threat was successful and she was promptly sent to the hospital for treatment.

As long as local cultural beliefs encourages older people to endure chronic pains and compliant attitudes, as aforementioned, older people’s proactive health-seeking behavior becomes a problem. The mother’s persistence with health-seeking and longevity was viewed as morally unacceptable by her family and the public. Her third son commented, asserting this as “unreasonable” and said it was “because she (the mother) was too scared of death. She had seen her 83rd already. Why can’t she just let it go?”. The view of “unreasonable” was also shared by other people in their circle who agreed that, as an older person, the mother had asked “too much” in the way of health-seeking, and it was her fault to pursue endless healthcare and longevity. For instance, another 80-year-old inpatient narrated her experiences and taught the mother to be “reasonable” in the face of family neglect: “Every time I was sick, my children just pretended that they had no idea about it. But that’s all right. Being an older person, you need to be content (“*zhizu*” in Chinese) with what you’ve got!”. 

Despite her resistance, the mother also internalized the public discouragement, expressing a deep concern of being accused as “immoral” and her family’s reluctance to take on the burden of support for her long-term health care: “If I continue living in the hospital, they will get very angry and stop offering me anything...” The result was that she attempted to cut short the length of her inpatient care, insisting on being discharged only after a few days of hospital treatment to return home for rehabilitation. Having experienced a very strained family relationship with her children after her suicide threat, it was unclear how she would be treated at home. 

## 4. Discussion 

Based on the narratives of older patients, their families, and other parties involved in seeking health care of older people, this paper adopts a relational account that focused on the interplay between the institutional structure and local cultural systems to examine issues of health-seeking of older people in a Chinese rural locality. A prevalent discouraging public attitude towards seeking the healthcare of older people was identified, which can be ascribed to the lack of access to necessary medical care under the structural constraint of the rural–urban divide as well as the unfavorable cultural values regarding seeking healthcare for the aged in China. Under the *hukou*-based rural–urban divide, older people in rural areas are deprived of access to healthcare, social pensions, and other aging support-related social welfare resources, thereby shifting the primary burden of aged healthcare onto individual families, impeded as they were by structural obstacles. Discouragement was further intensified by a lack of trust in the commodified healthcare industry in China, as well as the pessimism about the curability of and recovery from the chronic symptoms of older patients. Simultaneously, the local cultural values prioritizing personal characteristics such as endurance and self-restrained health-seeking behaviors have compounded this discouragement and devaluation. In order to become socially recognized self-restrained older people, most older patients strategically normalize chronic symptoms, actively avoiding seeking hospital care, while others adopt proactive health-seeking attitudes, to the contrary, which are perceived as culturally inappropriate.

## 5. Conclusions 

This paper highlights the values of and perspectives towards elderly health-seeking in an impoverished Chinese rural setting, of those older patients who are powerless and socio-economically disadvantaged, and whose voices have been largely omitted in the existing literature. An account of “positive” interpretations regarding older patients’ disengagement of health-seeking identified in the Chinese rural context is also in direct opposition to the conventional definition of “ideal” patients in the West, i.e., those who tend to be autonomous and who are able to proactively navigate institutional structures in existing health-seeking literature. This “positive” interpretation, however, does suggest that cultural devaluation of rural older members’ health-seeking is “accepted” from a standpoint of ethics or should be respected simply because of other people’s traditions. Rather, attention should be given to the powerless position and decreased gerontological status of rural older members when experiencing the hierarchical arrangement of the rural–urban structural divide and widespread ageist discriminations in China. In some sense, discouraging the seeking of care of older people mirrors the true predicament of health-seeking in rural regions and the public’s dissatisfaction with the current under-developed rural social welfare system in China. 

The focus of this paper on examining cultural forces pertinent to health-seeking is informative of the current health system reform in China. It sheds light on the need to mitigate both the structural and cultural impediments in advancing rural–urban health equity and promoting active health-seeking, as well as the need to understand health-seeking behavior as multidimensional rather than singularly determined by health resource distributions or insurance design. At the medical, institutional level, restructuring a supportive social welfare system that enables rural older populations who are at the bottom of the socio-economic ladder sufficient access to financial support in response to their healthcare needs is essential. Equally important is to eliminate discrimination against rural older members at the public perception level, and to increase the recognition regarding the significance of aging support within the whole society. 

This paper also has its limitations. Empirical data presented in this paper drew on ethnographical work conducted in only one hospital, which may not be entirely representative of the experiences of older patients in other rural regions in China. Despite this, the voices and experiences of older patients presented in this paper also point to a set of common predicaments and struggle that older people in other regions may also have encountered from accessing healthcare when experiencing socio-economic transformations. Many developing countries have a similar social backdrop comprising population aging, regional disparities in healthcare, and incomes, as well as widespread ageism and social discrimination over the process of social transitions. Therefore, findings reported in this study might also be informative to health-seeking issues in other developed countries.

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
