# Peer review of "“Unworthy of Care and Treatment”: Cultural Devaluation and Structural Constraints to Healthcare-Seeking for Older People in Rural China"

_ijerph, 2020, doi:10.3390/ijerph17062132_

Round 1
Reviewer 1 Report
I have carefully looked into the adaptations made to the manuscript and it is my opinion that it is now suitable for publication.
Author Response
Thank you!
Reviewer 2 Report
This research tried to clarify the present conditions of Chinese rural older people’s health care seeking behaviors based on social determinants of health. The idea is interesting, and the contents are clearly described.
The following suggestions are based on COREQ criteria.
- The research team and reflexivity
This research may not describe the interviewers, regarding their background, gender, professions, relationships with interviewees, and experiences of interviewing in qualitative research. These kinds of information are vital for confirmability. The authors can add them.
- Study design
-The author can also describe data saturation and member checking.
- Analysis and findings
- The contents are interesting and novel.
- discussion and conclusion
The contents are interesting and novel.
Author Response
I've revised the manuscript according to the comments (see the first two paragraphs in the subsection on "Methods and Materials" on p.5)This manuscript is a resubmission of an earlier submission. The following is a list of the peer review reports and author responses from that submission.
Round 1
Reviewer 1 Report
General recommendation:
This is a really interesting and timely research project on the politics of rural health care and the elderly in contemporary China, and it is great that the project is grounded in ethnographic research. Great job on tracing out the various discourse and their embedded contradictions. I strongly support the publication of the article, though it will be important that it be carefully and extensively edited by a native English speaker for syntax prior to publication--to ensure both the clarity of the arguments and the full academic respect that the research findings merit.
Here is my feedback on each of the sections of the article manuscript:
Abstract:
Great abstract. Though edit for English syntax. Probably replace “decimation” (line 25) with another term.
Introduction:
Good framing of significance of rural-urban health care disparity in contemporary China, and the particular contradictions this presents vis Confucian filial piety discourses vis the rural elderly.
1.1 “Backdrops” should probably be reframed as either “background” or “backdrop”
Good overview of the hukou system in first paragraph
Paragraph 2: you need to historically nuance this argument about modernization, urbanization, and discourses of self-reliance a bit more. During the Maoist period, rural health care (and economic well-being in general) was really premised on the idea of self-sufficiency/ self-reliance for the peasants/ rural China as well. The difference was that peasants also had access to town/ urban-based secondary and tertiary care that was covered by the state. Remember that the Maoist period was also about “modernization” in rural as well as urban contexts, albeit a narrative of “modernization” with different contours than subsequent, post-Mao narratives.
Good connection of the unworthiness/ worthlessness discourses (approx. lines 77-84) linked to economic shifts in the current era, though I think it is important as well as helpful to the reader to specify the actual Chinese terms deployed in this discourse here (as you do for other discourses later in the paper).
Paragraph 3 (and continuing into 4): last two sentences of para 3 (lines 97-100): be specific about time periods: Cooperative Medicine (hezuo yiliao)—at least the barefoot doctors narrative and practices—was really established at the beginning of the Cultural Revolution, and gets (mostly) disbanded with decollectivization (1981-3). While there are experimentations with health insurance schemes throughout rural China post decolllectivization, New Cooperative Medicine doesn’t really happen until 2009, right? Which is at least in good measure a response to the SARS epidemic in 2002-3? The radical for-profit entrepreneurialization of the urban hospital sector of course happened by the late 1990s.
1.2 Conceptual Framework: nice job on this section! Though of course in contemporary medical anthropology the general going assumption is that you simultaneously utilize both structural, political economy approaches and cultural, poststructuralist discursive approaches (i.e., vis China embedded in state-institution policy generated discourses). Syntax edits important in this section as well.
2. Materials and Methods: Good section! Review and edit carefully for syntax.
3. Results:
Nice job in this section at pulling out the three threads of discursive practice reflected in sections 3.1, 3.2, and 3.3. Really interesting findings and sets of contradictions! Will look forward to the broader dissertation findings. Though, again, vis 3.2 title—best to come up with an alternative to “decimation.”
Lines 262 & 263: again, it is important to include the actual Chinese terms & phrases for “worthlessness” and “unworthy of care & treatment.”
4. Discussion and Conclusions:
Lines 425-6: maybe use “impoverished” rather than “under-developed”
Second paragraph (especially sentences 2 & 3): nuance this argument about the rural Chinese case vis other models; note that the term “Western setting” really whitewashes over the incredible variability in forms of health care systems in the various nation-state projects that constitute “the West”.
Author Response
Dear review,
We thank you very much for your recommendation for the publication of this manuscript, as well as your suggestions and comments which are extremely helpful for improving our manuscript. Here are changes we have made accordingly:
Abstract: Great abstract. Though edit for English syntax. Probably replace “decimation” (line 25) with another term.
Response: the language has been edited professionally. We have replaced "decimation" with "devaluation".
1.1 “Backdrops” should probably be reframed as either “background” or “backdrop”
Response: the title has been corrected.
Paragraph 2: you need to historically nuance this argument about modernization, urbanization, and discourses of self-reliance a bit more. During the Maoist period, rural health care (and economic well-being in general) was really premised on the idea of self-sufficiency/ self-reliance for the peasants/ rural China as well. The difference was that peasants also had access to town/ urban-based secondary and tertiary care that was covered by the state. Remember that the Maoist period was also about “modernization” in rural as well as urban contexts, albeit a narrative of “modernization” with different contours than subsequent, post-Mao narratives.
Response: we have restructured the backdrop section with following changes:
- more information about comparison of health care systems under Maoist era and post-Maoist era.
- more inputs about the impact of modernisation on family support
- more discussion about a self-reliant social discourse and the fact that older people already been partially self-reliant prior to economy reforms.
Good connection of the unworthiness/ worthlessness discourses (approx. lines 77-84) linked to economic shifts in the current era, though I think it is important as well as helpful to the reader to specify the actual Chinese terms deployed in this discourse here (as you do for other discourses later in the paper).
Response: Related to the term “worthlessness”, the actual Chinese term we deployed is “buzhide zhi” that has been added in the manuscript as appropriate.
Paragraph 3 (and continuing into 4): last two sentences of para 3 (lines 97-100): be specific about time periods: Cooperative Medicine (hezuo yiliao)—at least the barefoot doctors narrative and practices—was really established at the beginning of the Cultural Revolution, and gets (mostly) disbanded with decollectivization (1981-3). While there are experimentations with health insurance schemes throughout rural China post decolllectivization, New Cooperative Medicine doesn’t really happen until 2009, right? Which is at least in good measure a response to the SARS epidemic in 2002-3? The radical for-profit entrepreneurialization of the urban hospital sector of course happened by the late 1990s.
Response: Following the reviewer’s suggestions, we have made three main changes in this section:
- More information about the community-funded Cooperative Medical Scheme and barefoot doctors prior to de-collectivization.
- Clarification of the implementation of universal health care coverage: it started in 2002, with over 90% rural population insured under the New Rural Cooperative Medical Scheme to 2009.
- The correction of time period regarding the enterprising of public hospitals.
1.2 Conceptual Framework: nice job on this section! Though of course in contemporary medical anthropology the general going assumption is that you simultaneously utilize both structural, political economy approaches and cultural, poststructuralist discursive approaches (i.e., vis China embedded in state-institution policy generated discourses). Syntax edits important in this section as well.
Response: the language has been edited.
- Materials and Methods: Good section! Review and edit carefully for syntax.
Response: a careful review and editing work has been done.
- Results:
Nice job in this section at pulling out the three threads of discursive practice reflected in sections 3.1, 3.2, and 3.3. Really interesting findings and sets of contradictions! Will look forward to the broader dissertation findings. Though, again, vis 3.2 title—best to come up with an alternative to “decimation.”
Lines 262 & 263: again, it is important to include the actual Chinese terms & phrases for “worthlessness” and “unworthy of care & treatment.”
Response: we have replaced “decimation” with “subordination”. The Chinese term and phrases we added in in correspondence with “worthlessness” and “unworthy of treament” are “buzhide” and “buzhide zhi”.
- Discussion and Conclusions:
Lines 425-6: maybe use “impoverished” rather than “under-developed”
Response: has been corrected
Second paragraph (especially sentences 2 & 3): nuance this argument about the rural Chinese case vis other models; note that the term “Western setting” really whitewashes over the incredible variability in forms of health care systems in the various nation-state projects that constitute “the West”.
Response: we have corrected the wording "Western setting".
Lastly, the language of this manuscript has been professionally edited.
Again, thank you for your time for reviewing our manuscript and positive responses and suggestions for improving it.
With best wishes,
Xiang
Reviewer 2 Report
This is a very interesting study, which points out an innovative topic about Chinese older people’s health seeking behavior. I have read the paper with a lot of attention and the findings are very important to policy makers and caregivers.
Hereby you can find some comments. I think it is very important that the English native authors read again the paper to address language issues. Some of the issues I list here:
1- ‘ This discouraged attitude was expressed by older people’s families and the public in a way that by devaluing older members’ health as “unworthy of care and treatment”.’
2- ‘these discouragement and devaluation’ – this
3- in accessing to health care - in accessing health care
4- Yet, recent empirical work suggests that reformative efforts at the institutional level have not been resulted in an obvious increase of medical care utilisation and health.
I would suggest ‘have not resulted in’
5- Such paradox suggests the needs for more research on what factors and constraints discourage the health-seeking behaviors of rural Chinese older populations. I would suggest the need’
6- An empirical exploration on the lived experiences of health-seeking of older people in rural China is also with potential to rectify this idealized assumption. I would suggest ‘has the potential of rectifying’
7- Additionally, modernization also alters the way people perceived the meaning of 76 late-age care and support.Different time tenses of the verbs.
8- 90% of rural dwellers. I would change for ‘older people from the rural area’.
9- It is reported that the proportion of households that endured catastrophic health expenditure and medical impoverishment in rural China as 14.4% and 9.2% respectively, which were higher than those found in urban China [2,26]. - Difficult to understand, please rephrase.
10- In Yan Long’s and Lydia Li’s study on rural-urban dichotomy in health-seeking behaviors in China, they summaries the various forces and constraints that contribute to the dichotomy which can be divided as structure-constrained and culture-determined[4].
11- Those local beliefs and values are already in place, which in turn hinder or facilities their decisions regarding whether and how to engage health care systems [33,36].
12- The structural and cultural facets intertwine in a way that produces inequalities, as coined in Long’s work, ”deeply imbued with legitimizing rationales wherein present prevailing
13-Too much repetition in this sentence:
Meanwhile, the voices and storytelling of other parties who overlapped with the older patients’ care and treatment, i.e. doctors, nurses, other patients and their family, and some family-paid caregivers, were also taken into accounts to formulate a nuanced picture of seeking health care in given family context. These multiple voices are helpful to form a nuanced picture of supporting health-seeking and care-giving in given family context.
14- A daughter-in-law who was accompany her mother-in-law for inpatient care due to cardiovascular problem described:
15- Please, in both cases, say that the person HAD or SUFFERED a stroke instead of using the word ‘attack’; Thank you.
For example, there was a 62-year-old participant who was attacked by a stroke and fainted over on the farmland.
Until he was re-attacked by another stroke and gradually became paralyzed was he re-enrolled for inpatient care in Qincun Hospital during the time of the current study.
16- Methods: Did you use any particular software in your analysis? If yes, please specify.
17- Please rephrase this sentence:
Only she felt so ill and even couldn’t stand still would she ask her son to send her to hospital for inpatient care.
18-
Mistakes in title: 3.3. Active resistance and expereinced moral acussation
19- Despite this, the voices and experiences of older patients presented in this paper also points to a set of common predicaments and struggles that older people in other regions may also encountered from accessing health care when experiencing socio- economic transformations.
May also HAVE.
20- Many developing countries encounter a similar social backdrops comprising population ageing, regional disparities in health care and incomes, and widespread ageism and social discrimination over the process of social transitions.
A similar is single; backdrops is plural!
21- Mistake in the reference: World health bckground paper: Helath insurance
Author Response
Dear reviewer,
We thank you very much for your careful review with suggestions and comments for improving our manuscript. Here are major changes we have made:
Thank you for the syntax errors you have kindly pointed out in 1-15 and 17-21 comments, we have addressed them accordingly. Meanwhile, the language of the whole manuscript has been professionally edited.
Responding to your 16th comment: We made further elaboration of our research methods. The software we used for analysing data is Atlas. ti., we have addressed it properly.
Again, thank you for your careful review with positive comments.
Best wishes,
Xiang